# TropEx: An Algorithm for Extracting Linear Terms in Deep Neural Networks

**Martin Trimmel**[*1]**, Henning Petzka**[*1]**, Cristian Sminchisescu**[1,2]
[1]Lund University    [2]Google Research
{martin.trimmel, henning.petzka, cristian.sminchisescu}@math.lth.se

## Abstract

Deep neural networks with rectified linear (ReLU) activations are piecewise linear functions, where hyperplanes partition the input space into an astronomically high number of linear regions. Previous work focused on counting linear regions to measure the network's expressive power and on analyzing geometric properties of the hyperplane configurations. In contrast, we aim to understand the impact of the linear terms on network performance, by examining the information encoded in their coefficients. To this end, we derive TropEx, a non-trivial tropical algebra-inspired algorithm to systematically extract linear terms based on data. Applied to convolutional and fully-connected networks, our algorithm uncovers significant differences in how the different networks utilize linear regions for generalization. This underlines the importance of systematic linear term exploration, to better understand generalization in neural networks trained with complex data sets.

## 1 Introduction

Many of the most widely used neural network architectures, including VGG (Simonyan & Zisserman, 2015), GoogLeNet (Szegedy et al., 2015) and ResNet (He et al., 2016), make use of rectified linear activations (ReLU, (Hahnloser et al., 2000; Glorot et al., 2011), i.e., $\sigma(x) = \max\{x, 0\}$) and are therefore piecewise linear functions. Despite the apparent simplicity of these functions, there is a lack of theoretical understanding of the factors that contribute to the success of such architectures. Previous attempts of understanding piecewise linear network functions have focused on estimating the number of **linear terms**, which are the linear pieces (affine functions) that constitute the network function. A **linear region** is being defined as a maximally connected subset of the input space on which the network function is linear. Since computing the exact number of linear regions is intractable, work has focused on obtaining upper and lower bounds for this number (Arora et al., 2016; Serra et al., 2018; Pascanu et al., 2013; Raghu et al., 2017; Montufar et al., 2014; Montúfar, 2017; Xiong et al., 2020; Zhang et al., 2018). To our knowledge, the currently best upper and lower bounds were calculated by Serra et al. (2018). Raghu et al. (2017) show these bounds to be asymptotically tight.

All of the mentioned papers share the intuition that the number of linear regions of neural networks measures their expressivity. Since the bounds grow linearly in width and exponentially in depth, deep networks are interpreted to have greater representational power. However, these bounds are staggeringly high: the upper bound on the number of linear regions in (Serra et al., 2018) exceeds $10^{300}$ even for the smallest networks we experimented on. (There are approximately $10^{80}$ atoms in the universe.) For slightly larger networks, the upper bound exceeds $10^{17000}$ whereas the lower bound exceeds $10^{83}$ linear regions. The number of training samples is generally much smaller than the estimated number of linear regions ($\leq 10^6$), so that almost none of the linear regions contains training data. This raises the question of how representative the number of linear regions is for network performance and how information extracted from training samples passes on to the many linear regions free of data for successful generalization to test data.

There are indications that a high number of linear regions is not required for good network performance. Frankle & Carbin (2019) point out that smaller networks perform similarly well as large ones, when a suitable initialization of the smaller network can be found from training the larger one. Hence,

---

*Denotes equal contribution.

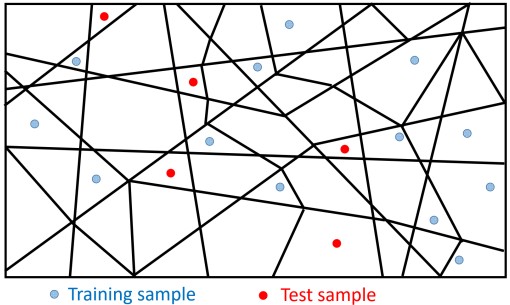 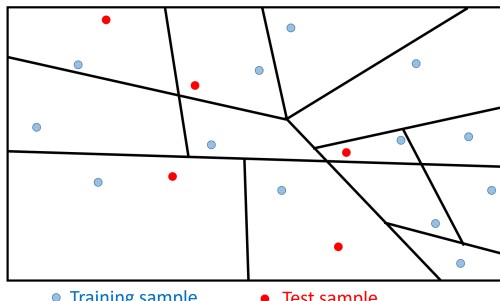

Figure 1: A ReLU network function before *(left)* and after *(right)* extraction. *Left*: Hyperplanes separate the input space into linear regions. Most of them do not contain any data points. Each data point occupies its own linear region. *Right:* After extraction, the function remains unchanged on the linear regions of training samples. Test samples now fall into regions of training samples.

the expressivity of the large network is helpful to explore the parameter space, but the small, less expressive network is sufficient to achieve high accuracy. Lee et al. (2019) and Croce et al. (2018) modify the training loss to encourage larger linear regions with the goal of robustness to adversarial attacks. Hanin & Rolnick (2019b;a) argue that in practice there are fewer linear regions than expected from the bounds and empirically investigate this for the MNIST data set. All these observations question the explanatory power of astronomically high bounds for the number of linear regions. More recently, the focus of research on linear regions has been shifting away from pure counting towards an understanding of the linear regions themselves. Zhang & Wu (2020) study geometric properties of linear regions and notice that batch normalization and dropout, albeit leading to similar network accuracies, produce differently looking linear regions.

Our approach to the understanding of linear regions differs in that it investigates the linear coefficients of linear regions. To this end, we propose **TropEx**, a tropical algebra-based algorithm extracting linear terms of the network function $\mathcal{N}$ (Figure 1) using a data set $\mathcal{X}$. **TropEx outputs an extracted function $\mathcal{N}^{(\mathcal{X})}$ containing only the linear terms corresponding to regions on which data lies.** As a result, $\mathcal{N}$ and $\mathcal{N}^{(\mathcal{X})}$ agree on neighbourhoods of all data points. This creates a tool for the study of generalization from a new viewpoint, i.e., the perspective of linear regions and their coefficients.

**Our contributions are as follows:**
• A new computational framework representing tropical functions (Definition B.4) as matrices to efficiently perform tropical calculations appearing in networks with rectified linear activations.
• This framework allows us to derive TropEx, an algorithm to systematically extract linear terms from piecewise linear network functions.[1]
• An application of TropEx to fully-connected (FCN) and convolutional networks (CNN) reveals that (i) consistently all training and test samples fall into different linear regions; (ii) Simple tasks (MNIST) can be solved with the few linear regions of training samples alone, while this does not hold for more complex data sets. (iii) FCNs and CNNs differ in how they use linear regions free of training data for their performance on test data: Several measures illustrate that CNNs, in contrast to FCNs, tend to learn more diverse linear terms. (iv) We confirm that the number of linear regions alone is not a good indicator for network performance and show that the coefficients of linear regions contain information on architecture and classification performance.

## 2 BACKGROUND AND OVERVIEW

It was recently shown by Charisopoulos & Maragos (2018); Zhang et al. (2018) that ReLU neural network functions are the same as *tropical rational maps*. Tropical rational maps are exactly those functions where each entry in the output vector can be written as a difference of maxima

$$\mathcal{N}_i(\mathbf{x}) = \max\{a_{i1}^+(\mathbf{x}), \ldots, a_{in}^+(\mathbf{x})\} - \max\{a_{i1}^-(\mathbf{x}), \ldots, a_{im}^-(\mathbf{x})\}, \tag{1}$$

---

[1]Link to open source implementation: `https://github.com/martrim/tropex`

where each $a_{ij}^+, a_{ij}^- : \mathbb{R}^d \to \mathbb{R}$ is an affine function with only positive coefficients, taking the form $\mathbf{x} \mapsto \sum_j w_j x_j + w_0$ with all $w_j \in \mathbb{R}_{\geq 0}$. Since the number of terms in (1) dwarfs the number of atoms in the universe, it is impossible to obtain this expression in practice. Therefore, we only extract those terms that correspond to linear regions of data points. For a fixed data point $\mathbf{x} \in \mathcal{X}$, the maximum of the network outputs can be written as $\max_i \mathcal{N}_i(\mathbf{x}) = a_{\mathbf{x}}^+(\mathbf{x}) - a_{\mathbf{x}}^-(\mathbf{x})$, where $a_{\mathbf{x}}^+, a_{\mathbf{x}}^-$ are the affine functions such that $a_{\mathbf{x}}^+(\mathbf{x}) \geq a_{ij}^+(\mathbf{x}), a_{\mathbf{x}}^-(\mathbf{x}) \geq a_{ij}^-(\mathbf{x})$ for all $i, j$. TropEx extracts $a_{\mathbf{x}}^+$ and $a_{\mathbf{x}}^-$. The extracted terms can be used to construct a tropical map $\mathcal{N}^{(\mathcal{X})}(\mathbf{x}) = \left(\mathcal{N}_1^{(\mathcal{X})}(\mathbf{x}), \ldots, \mathcal{N}_s^{(\mathcal{X})}(\mathbf{x})\right)$ with maximally enlarged linear regions, given by

$$\mathcal{N}_i^{(\mathcal{X})}(\mathbf{x}) = \max\{a_{\mathbf{x}_{k_1}}^+(\mathbf{x}), \ldots, a_{\mathbf{x}_{k_{D_i}}}^+(\mathbf{x})\} - \max\{a_{\mathbf{x}_{k_1}}^-(\mathbf{x}), \ldots, a_{\mathbf{x}_{k_{D_i}}}^-(\mathbf{x})\}, \tag{2}$$

where there are $D_i$ data points $\mathbf{x}_{k_1}, \ldots, \mathbf{x}_{k_{D_i}}$ given label $i$ by the original network. Being a tropical rational map, the function $\mathcal{N}^{(\mathcal{X})}$ is again a ReLU neural network function by Zhang et al. (2018). The maximal entries of the two output vectors (hence also the assigned labels) of the extracted function $\mathcal{N}^{(\mathcal{X})}$ and the original network $\mathcal{N}$ agree in the neighbourhood of any data point $\mathbf{x} \in \mathcal{X}$.

We discuss the basics of tropical algebra in Appendix B.1 and refer to Maclagan & Sturmfels (2015) for a detailed introduction. The relation of tropical geometry and ReLU networks is studied in Zhang et al. (2018); Charisopoulos & Maragos (2018); Alfarra et al. (2021).

## 3 METHOD

### 3.1 MATRIX REPRESENTATION OF TROPICAL RATIONAL MAPS

If one were to represent tropical rational maps symbolically on a computer, computations would be too slow. Therefore, we present tropical rational maps as multi-dimensional arrays.

**Definition 3.1.** Given an affine function $a : \mathbb{R}^{d_0} \to \mathbb{R}; \mathbf{x} \mapsto \sum_k w_k x_k + w_0$, we will call the vector $(w_0, w_1, \ldots, w_{d_0})$ its *coefficient vector*, the scalar $w_0$ its *constant part* and the vector $(w_1, \ldots, w_{d_0})$ its *variable part*.

We can represent functions $\mathcal{N}_i : \mathbb{R}^{d_0} \to \mathbb{R}$ as in equation (1) in the following way: Let the rows of the matrix $\mathbf{A}_i^+ \in \mathbb{R}^{n \times d_0}$ and the vector $\mathbf{a}_i^+ \in \mathbb{R}^{n \times 1}$ be the variable and the constant parts of the affine functions $a_{ij}^+$, respectively. (Analogously for $\mathbf{A}_i^-$ and $\mathbf{a}_i^-$.) We can then define $(\mathbf{A}_i^+, \mathbf{a}_i^+)(\mathbf{x}) = \max\{\mathbf{A}_i^+ \mathbf{x} + \mathbf{a}_i^+\}$, where the maximum is taken over the rows of the resulting column vector. If we define the formal quotient[2] of matrix-vector pairs by $(\mathbf{A}_i^+, \mathbf{a}_i^+)/(\mathbf{A}_i^-, \mathbf{a}_i^-)(\mathbf{x}) = \max\{\mathbf{A}_i^+ \mathbf{x} + \mathbf{a}_i^+\} - \max\{\mathbf{A}_i^- \mathbf{x} + \mathbf{a}_i^-\}$, then $\mathcal{N}_i(\mathbf{x}) = (\mathbf{A}_i^+, \mathbf{a}_i^+)/(\mathbf{A}_i^-, \mathbf{a}_i^-)(\mathbf{x})$, giving us a matrix-representation of the function $\mathcal{N}_i$. An entire network function with $s$ output dimensions can then be represented by a list $\left((\mathbf{A}_i^+, \mathbf{a}_i^+)/(\mathbf{A}_i^-, \mathbf{a}_i^-)\right)_{1 \leq i \leq s}$.

The advantage of the proposed matrix representation of tropical rational maps are natural operations performing calculations that arise for (concatenations of) layers of neural networks (see supplements). A dense layer $\ell : \mathbb{R}^{d_1} \to \mathbb{R}^{d_2}$ with ReLU activation is represented as a list $\left((\mathbf{A}_i^+, \mathbf{a}_i^+)/(\mathbf{A}_i^-, \mathbf{a}_i^-)\right)_{1 \leq i \leq d_2}$. Denoting by $\mathbf{W}^{\text{pos}}$ and $\mathbf{W}^{\text{neg}}$ the positive and negative part of a matrix $\mathbf{W}$, respectively, i.e. $w_{ij}^{\text{pos}} = \max\{w_{ij}, 0\}$ and $w_{ij}^{\text{neg}} = \max\{-w_{ij}, 0\}$, the matrix representation of a single neuron $n_i(\mathbf{x}) = \max\{\mathbf{w} \cdot \mathbf{x} + b, 0\}$ is given by

$$\mathbf{A}_i^+ = \begin{pmatrix} \mathbf{w}^{\text{pos}} \\ \mathbf{w}^{\text{neg}} \end{pmatrix}, \ \mathbf{a}_i^+ = \begin{pmatrix} b^{\text{pos}} \\ b^{\text{neg}} \end{pmatrix}; \quad \mathbf{A}_i^- = \mathbf{w}^{\text{neg}}, \ \mathbf{a}_i^- = b^{\text{neg}}.$$

### 3.2 EXTRACTING LINEAR TERMS OF A CLASSIFICATION NETWORK

We now consider a classification neural network $\mathcal{N}$ with $s$ labels. We show that we can represent the network $\mathcal{N}$ with a matrix-vector pair $(\mathbf{A}^-, \mathbf{a}^-)$ in the denominator that is constant over all output dimensions. The proof is given in Section C.2 of the supplementary material.

---

[2]This is in line with the tropical algebra notation: a *tropical quotient* is the same as a usual difference.

---

**Algorithm 3.1** TropEx: Extracting Linear Terms of a Neural Network

---

**Inputs**:    Neural Network $\mathcal{N}$
                Data set $\mathcal{X} = \{(\mathbf{x}_{i_k}, i)\}$ with $D_i$ points of label $i$
**Output**:   Extracted Function $\mathcal{N}^{(\mathcal{X})} = ((\mathbf{A}_i^+, \mathbf{a}_i^+)/(\mathbf{A}^-, \mathbf{a}^-))_{1 \leq i \leq s}$

 1: $\mathbf{W}, \mathbf{b} \leftarrow$ weight matrix and bias vector of last layer $\ell$
 2: $\mathbf{C}^-, \mathbf{c}^- \leftarrow$ column sums of $\mathbf{W}^{\text{neg}}, \mathbf{b}^{\text{neg}}$
 3: $\mathbf{C}^+ \leftarrow \mathbf{W} + \mathbf{C}^-, \mathbf{c}^+ \leftarrow \mathbf{b} + \mathbf{c}^-$
 4: **for** $i = 1$ to $s$ **do**
 5:      $\mathbf{A}_i^+ \leftarrow \text{rep}(\mathbf{C}_{i\bullet}^+, D_i), \mathbf{a}_i^+ \leftarrow \text{rep}(\mathbf{c}_i^+, D_i)$         ▷ Repetition $D_i$ times along the rows.
 6: $\mathbf{A}^- \leftarrow \text{rep}(\mathbf{C}^-, D), \mathbf{a}^- \leftarrow \text{rep}(\mathbf{c}^-, D)$         ▷ $D$ = total no of data points
 7: $\mathbf{A}_{\max} \leftarrow$ maxima of the columns of all $\mathbf{A}_i^+$ and $\mathbf{A}^-$ stacked
 8: **for** last layer $\ell$ in $\mathcal{N}$ not yet used **do**
 9:      $(\mathcal{N}^{(\mathcal{X})}, \mathbf{A}_{\max}) \leftarrow \text{merge}_\ell (\mathcal{N}^{(\mathcal{X})}, \mathbf{A}_{\max})$ according to Table 1

---

**Lemma 3.2.** *Let $\mathcal{N} : \mathbb{R}^d \to \mathbb{R}^s$ be the function of a ReLU neural network for classification with $s$ output neurons. Then there are affine functions $a_{ij}^+, a_j^-$ such that*

$$\mathcal{N}(\mathbf{x}) = \begin{pmatrix} \max\{a_{11}^+(\mathbf{x}), \ldots, a_{1n_1}^+(\mathbf{x})\} \\ \vdots \\ \max\{a_{s1}^+(\mathbf{x}), \ldots, a_{sn_s}^+(\mathbf{x})\} \end{pmatrix} - \max\{a_1^-(\mathbf{x}), \ldots, a_m^-(\mathbf{x})\}, \tag{3}$$

*where the maxima of the $a_i^-(\mathbf{x})$ on the right is subtracted from each entry of the vector on the left.[3] In terms of our matrices the classification network $\mathcal{N}$ with $s$ labels can be represented by a list $((\mathbf{A}_i^+, \mathbf{a}_i^+)/(\mathbf{A}^-, \mathbf{a}^-))_{1 \leq i \leq s}$ of matrix-vector pairs. The label is then given by $\text{argmax}_{1 \leq i \leq s} (\mathbf{A}_i^+, \mathbf{a}_i^+)(\mathbf{x})$.*

**How to get the extracted function $\mathcal{N}^{(\mathcal{X})}$ of (2) from the network $\mathcal{N}$.** TropEx extracts, for each data point $\mathbf{x}_k$ of label $i$, the affine functions $a_{ij}^+$ and $a_l^-$ from the network representation in (3) such that $a_{ij}^+(\mathbf{x}_k) \geq a_{i\tilde{j}}^+(\mathbf{x}_k), a_l^-(\mathbf{x}_k) \geq a_{\tilde{l}}^-(\mathbf{x}_k)$ for all $\tilde{i}, \tilde{j}, \tilde{l}$. We start with the last layer of the network and inductively merge new layers into the existing matrix pairs. The merge operations depend on the type of the layer as shown in Table 1. Putting things together gives Algorithm 3.1, which has the extracted function $\mathcal{N}^{(\mathcal{X})}$ as its output. The run-time and storage complexities per data point correspond to 3 forward passes through the network. Theorem 3.3 states that TropEx indeed results in a selection of linear terms based on a data set of points and that the extracted tropical function agrees with the network on neighbourhoods of all these points. Its proof and the complete, non-trivial derivation of the algorithm are in the appendix, where we develop a framework that enables calculations on tropical matrices that correspond to manipulations of the tropical functions they represent. For illustrative purposes, we also present a worked-out example of applying TropEx to a toy neural network there.

**Theorem 3.3.** *Let $\mathcal{N} = (\mathcal{N}_1, \ldots, \mathcal{N}_s) : \mathbb{R}^d \to \mathbb{R}^s$ be the function of a ReLU neural network for classification into $s$ classes. Let $\mathcal{N}^{(\mathcal{X})}$ be the network obtained from Algorithm 3.1, applied to $\mathcal{N}$ using a data set $\mathcal{X} = \{(\mathbf{x}_{k_j}, i) | 1 \leq i \leq s, 1 \leq j \leq D_i\}$ of $D_i$ data points $\mathbf{x}_{k_j}$ given label $i$ by $\mathcal{N}$. (There are $D$ points $\mathbf{x}_1, \ldots, \mathbf{x}_D$ in total). Then, (1) for all labels $i$, $\mathcal{N}_i^{(\mathcal{X})}(\mathbf{x}) = \max\{a_{i1}^+(\mathbf{x}), \ldots, a_{iD_i}^+(\mathbf{x})\} - \max\{a_1^-(\mathbf{x}), \ldots, a_D^-(\mathbf{x})\}$, where the $a_{ij}^+$ and $a_l^-$ are extracted from a representation of $\mathcal{N}_i$ as in Equation 3; and (2) every data point $\mathbf{x}_k$ has a neighbourhood $U_k$ on which the maximum of the extracted function agrees with the maximal network output:*

$$\max_{1 \leq i \leq s} \mathcal{N}_i^{(\mathcal{X})}(\mathbf{x}) = \max_{1 \leq i \leq s} \mathcal{N}_i(\mathbf{x}) \quad \text{for all } \mathbf{x} \in U_k.$$

*In particular, $\mathcal{N}^{(\mathcal{X})}$ and $\mathcal{N}$ classify all points in $U_k$ by assigning the same label.*

---

[3]This implies that every ReLU neural network classifier can be represented by a convex function.

| Type | Operation | Type | Operation |
|------|-----------|------|-----------|
| BNorm | $\gamma, \beta, \mu, \sigma, \epsilon \leftarrow$ Batchnorm parameters
$s \leftarrow \gamma/\sqrt{\sigma^2 + \epsilon}, t \leftarrow -\mu \cdot s + \beta$
$\mathbf{A} \leftarrow s\mathbf{A}; \mathbf{a} \leftarrow \mathbf{a} + \mathbf{A}t$
$\mathbf{A}_{\max} \leftarrow |s|\mathbf{A}_{\max}$ | Maxpool | $\mathbf{A}, \mathbf{A}_{\max} \leftarrow$ repeat to
input shape of $\ell$
$\mathbf{A}_k \leftarrow$ set 0 according to
activations of $\bar{\ell}(\mathbf{x}_k)$ |
| Conv | $F, \mathbf{b} \leftarrow$ filter, bias of $\ell$
$K \leftarrow \text{ConvTrans}(\mathbf{A}_{\max}, F^{\text{neg}})$
$\mathbf{A}_{\max} \leftarrow K + \text{ConvTrans}(\mathbf{A}_{\max}, F^{\text{pos}})$
$\mathbf{a} \leftarrow \mathbf{a} + \mathbf{A}\mathbf{b}$
$\mathbf{A} \leftarrow K + \text{ConvTrans}(\mathbf{A}, F)$ | Dense | $\mathbf{W}, \mathbf{b} \leftarrow$ weights, bias of $\ell$
$K \leftarrow \mathbf{A}_{\max}\mathbf{W}^{\text{neg}}$
$\mathbf{A}_{\max} \leftarrow K + \mathbf{A}_{\max}\mathbf{W}^{\text{pos}}$
$\mathbf{a} \leftarrow \mathbf{a} + \mathbf{A}\mathbf{b}$
$\mathbf{A} \leftarrow K + \mathbf{A}\mathbf{W}$ |
| Flatten | $\mathbf{A}, \mathbf{A}_{\max} \leftarrow$ reshape to input shape of $\ell$ | L-ReLU | $\alpha \leftarrow$ Leaky ReLU parameter |
| ReLU | $\mathbf{A}_{kj} \leftarrow 0$ if $\bar{\ell}(\mathbf{x}_k)_j = 0$ | | $\mathbf{A}_{kj} \leftarrow \alpha \cdot \mathbf{A}_{kj}$ if $\bar{\ell}(\mathbf{x}_k)_j \leq 0$ |

Table 1: Merge operations $(\mathcal{N}^{(\mathcal{X})}, \mathbf{A}_{\max}) \mapsto (\mathcal{N}^{(\mathcal{X})}, \mathbf{A}_{\max})$, according to type of layer $\ell$. $\mathbf{A}_k$ denotes the slice of $\mathbf{A}$ corresponding to data point $k$. $\bar{\ell}$ denotes all of the network up to and including $\ell$. Read expressions like $\mathbf{A} \leftarrow K + \mathbf{a}\mathbf{W}$ as $\mathbf{A}_i^+ \leftarrow K + \mathbf{A}_i^+\mathbf{W}; \mathbf{A}^- \leftarrow K + \mathbf{A}^-\mathbf{W}$ for all $i$.

## 4 EXPERIMENTS

TropEx extracts a function containing only linear terms corresponding to regions on which the given data lies. The extracted function agrees with the network on this data. This allows us to compare linear regions of train and test data, to separate the network structure from the information contained in the linear coefficients, and to test how well the linear coefficients generalize to test data.

**Setup** We train neural networks on MNIST (LeCun et al., 2010) and CIFAR10 (Krizhevsky, 2009). After training, we use training data points $\mathbf{x}^{(tr)}$ to extract linear terms $a_{\mathbf{x}^{(tr)}}^+(\mathbf{x})$ and $a_{\mathbf{x}^{(tr)}}^-(\mathbf{x})$ and construct an extracted function $\mathcal{N}^{(\mathcal{X})}$ as in equation (2). For some experiments, we also extract linear terms $a_{\mathbf{x}^{(te)}}^+(\mathbf{x})$ and $a_{\mathbf{x}^{(te)}}^-(\mathbf{x})$ corresponding to test data points $\mathbf{x}^{(te)}$. Regarding the architecture, we use fully-connected networks, AllCNN-C from Springenberg et al. (2015) and variations of VGG-B from Simonyan & Zisserman (2015). Section D in the appendix summarizes all architectures we used in our experiments. If not stated otherwise, we use architectures *Conv* for CNNs and *FCN8* for FCNs in our experiments. It is not our goal to train networks to state-of-the-art performance, but rather to compare variations of simple networks which are composed of the layers shown in Table 1. All layers have ReLU activations except for the last layer where we apply a softmax output function into the ten respective classes.[4] We train five networks of each architecture to ensure the consistency of our results. Further details on the training setup can be found in the appendix.

**Train and Test Linear Regions** At first, we investigate how training and test samples are distributed over the linear regions of the neural networks. For each data point $\mathbf{x}$, let $a_{\mathbf{x}} = a_{\mathbf{x}}^+ - a_{\mathbf{x}}^-$ be the function corresponding to the linear region on which $\mathbf{x}$ lies. Observing that $a_{\mathbf{x}} \neq a_{\mathbf{x}'}$ for all training and test points $\mathbf{x}, \mathbf{x}'$, we see that all points lie in different linear regions.[5] This is not a result of overfitting during training: All data points also occupy different regions when we check the linear regions after 1, 3, 5, 10, 15, 20, 30, 40, and 50 epochs of training on CIFAR10, and from epochs 1 to 20 on MNIST. Therefore we conclude that generalization capabilities of neural networks cannot be explained by test samples falling into the same linear region as training samples (or, in other words, by test samples inducing the same activation pattern as training samples).

**Examining Function Coefficients** With test samples falling into different regions than training samples, it is conceivable that neighboring regions could still be "nearly identical" and test samples would fall into such neighboring regions of training samples. To clarify, we ran experiments to test the similarity of linear regions for test samples before and after reduction. We take a test sample and extract the linear coefficient of its linear region in both the original network $\mathcal{N}$ and in the extracted function $\mathcal{N}^{(\mathcal{X})}$, where $\mathcal{X}$ is the training data. For these linear coefficient

---

[4]We also experimented with replacing ReLU with Leaky ReLU. Our observations are in line with what we describe for ReLU. More details can be found in the appendix, Section E.3.

[5]Except for the small 2-layer architecture, where the data points lie on 59,850 regions instead of 60,000.

| Name | $\mathcal{N}$ vs $\mathcal{N}^{(\mathcal{X})}$ | $\mathcal{N}$ vs true | $\mathcal{N}^{(\mathcal{X})}$ vs true |
|---|---|---|---|
| **FCN MNIST** | $97.7_{\pm 0.2}$ | $98.1_{\pm 0.2}$ | $97.6_{\pm 0.2}$ |
| **CNN MNIST** | $95.7_{\pm 0.3}$ | $99.2_{\pm 0.1}$ | $95.8_{\pm 0.2}$ |
| **FCN CIFAR10** | $52.5_{\pm 3.5}$ | $49.2_{\pm 0.7}$ | $38.2_{\pm 1.5}$ |
| **CNN CIFAR10** | $30.8_{\pm 1.3}$ | $71.1_{\pm 0.5}$ | $30.3_{\pm 1.4}$ |

Table 2: Results on test data after extraction of linear terms. Column 1: Agreement of network $\mathcal{N}$ with extracted function $\mathcal{N}^{(\mathcal{X})}$. Columns 2&3: Multi-class accuracy for $\mathcal{N}$ and $\mathcal{N}^{(\mathcal{X})}$. All values are averages over 5 runs and over each of the architectures in table 3 in section D of the appendix.

vectors, we calculate the (i) angle and (ii) Euclidean norm difference. Figure 2 shows both values for all test samples, where we differentiate between those test points $\mathbf{x}^{(te)}$ that get correctly classified by $\mathcal{N}^{(\mathcal{X})}$ (in blue) and those that get a wrong label (in red). We observe a clear difference between the CNN and the FCN. For CNNs, the coefficient vectors of both training and test affine functions are all close to orthogonal for correctly as well as incorrectly classified points, For FCNs, the angle and distance of correctly classified points are smaller than for incorrectly classified ones, but still far away from zero and therefore rule out the possibility of test samples falling into very similar neighboring regions. Finally, instead of comparing linear coefficients, we also tested the similarity in activation patterns before and after extraction. The results in section E.5 of the appendix show that in each layer approx $80\%$ of neuron activations agree between test and training region, showing that also the activation patterns of test samples deviate considerably and generalization cannot be simply explained by very similar activation patterns.

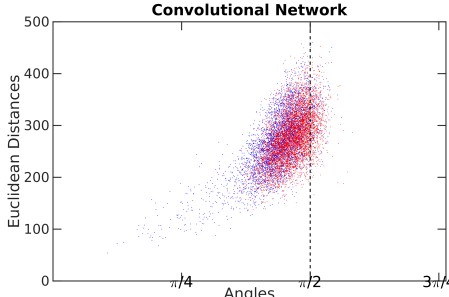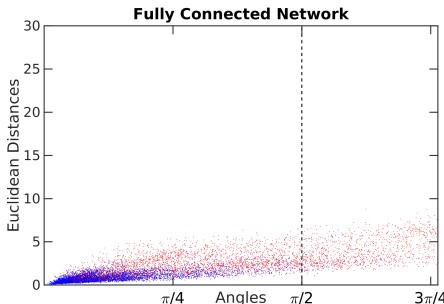

Figure 2: Angle and distance between coefficients of the linear regions of test data and training data used for correct (blue) and incorrect (red) classification by the extracted function. All angles for CNNs (left) are close to orthogonal, while FCNs (right) shows clear correlations between angles, distances and correctness of prediction.

**Accuracy of the Extracted Functions**    As predicted by Theorem 3.3, the maximum of the extracted function $\mathcal{N}^{(\mathcal{X})}$ agrees with the maximum of the original network on all training points for each of our networks. In particular, network and extracted function assign the same label to each training point. To investigate how well the coefficients of these linear regions generalize to unseen data, we compare test accuracy of network and extracted function in Table 2. We see a consistent difference between CNNs and FCNs across both data sets and all architectures: There is a drastic drop in the test accuracy of the CNNs, as opposed to a relatively small drop in the accuracy of the FCNs. Interestingly, for MNIST, the extracted tropical function has almost the same test accuracy as the original network. This is surprising as all known bounds on the number of linear regions of the original network suggest numbers of the order of $10^{80}$ up to over $10^{17000}$ from which we only observe 60.000 after reduction. Hence, for fully-connected networks on a simple task, the coefficients used on training data generalize well to test data, but for complex data, the learned coefficients generalize worse. This is remarkable, since previous studies (Hanin & Rolnick, 2019b;a; Zhang & Wu, 2020) of linear regions were forced to base their experiments on small data sets (or small networks) for computational reasons, and it seems that care must be taken when generalizing observations to more complex tasks. Moreover, the results reveal another difference between FCNs and CNNs that we further investigate.

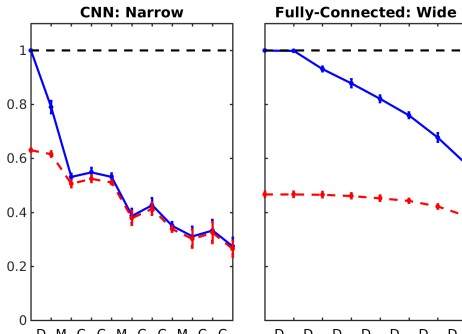 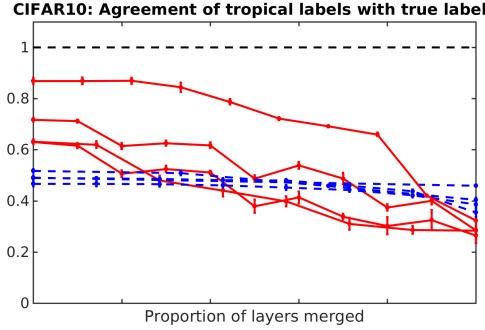

Figure 3: Left: Comparison of average test accuracy (dotted red) and agreement with labels assigned by the original network (blue) for networks *Narrow* and *Wide* while being transformed into tropical functions. The letters D,M,C denote dense, maxpooling and convolutional layers, respectively. Right: Average test accuracy for all CIFAR networks while being transformed into a tropical function. Fully-connected: dotted blue lines, CNNs: full red lines. The curves show a significant difference between the network types with FCN performance being more stable to extraction of linear regions.

**Transformation Process**    Figure 3, left shows how the test accuracy and agreement with the original network develop during the transformation from full network $\mathcal{N}$ to its extracted form $\mathcal{N}^{(\mathcal{X})}$. Starting with the last layer, TropEx iteratively merges layers into an extracted function. The x-axis shows which layers have been merged to an extracted tropical function at each step. The graph shows accuracy values for passing test samples through the original network until the layer that represents the input to the extracted tropical function and then applying this extracted function. At the right end of the plots, all layers have been merged to the tropical function $\mathcal{N}^{(\mathcal{X})}$. There is a clear difference between fully-connected networks and CNNs, which is consistent over all networks (Figure 3, right).

**Number of linear regions**    The fully-connected network *Wide* and the CNN *Narrow* have the same number of nodes after each parameter layer. Since *Narrow* has only few connections between its nodes and *Wide* is fully-connected, it is reasonable to assume that the number of linear regions of *Wide* is greater than the one of *Narrow* as its theoretical upper bound is higher. Hence, it would be expected that extraction of a fixed number of linear terms resulted in a smaller change of results for an initially worse performing *Narrow*, but the drop in test accuracy for *Narrow* is almost 5 times the drop for *Wide* (36.6% vs 8.1%). An estimate of the number of linear regions in practice (Appendix E.8) further suggests that *Narrow* has more linear regions than *Wide*, both being astronomically high. This all contradicts our intuition about how CNNs and FCNs work from the perspective of network expressivity in terms of bounding the maximal number of linear regions.

**Network Training**    We compare the performance of extracted functions $\mathcal{N}^{(\mathcal{X})}$ during the training of the network $\mathcal{N}$. Figure 4 displays the test accuracy (mean and standard deviation over 5 networks) of the extracted function and the agreement of label assignments of extracted form with the original network function for *Narrow* (CNN) and *Wide* (fully-connected) for several epochs. The difference between fully-connected and convolutional networks is here even more striking. For the CNN, the agreement between the extracted tropical function and the original network function falls rapidly after only one epoch and only slightly reduces from there. For the FCN, the agreement decreases slowly over the entire 50 training epochs and it never reaches a value as low as the CNN after its first epoch.

**Information encoded in linear coefficients**    The extracted functions all share the same number of linear terms, hence their difference in performance must be explained by the coefficient values. With this in mind, we attempt an interpretation of the above results and hypothesize that the difference lies in how FCN and CNN store important information for the classification task in the coefficients of linear regions. An FCN has the full freedom to compose weights to tailored coefficients of linear regions, whereas CNNs impose a structure on the weight space by filters and weight sharing, which results in the incapability to compose tailored linear coefficients for correct label assignments. Instead, the structural properties of convolutions play a significant role in generalization, which we remove by extracting linear terms. This changes the outcome on test data as the coefficients of linear regions

alone are limited in meaning. As training progresses, to achieve higher accuracy, the FCN reduces the information stored in linear coefficients and also learns to use some structure, so that the removal of this structure could explain the decrease in performance of the extracted function. An experiment, where we visually inspect misclassified images (Appendix E.2) is in line with this interpretation suggesting that the object shape is encoded in the linear coefficients of the FCNs, but for CNNs only simple features such as background color are encoded in the linear coefficients of linear regions. We visualize coefficients in E.7 to further support observed differences.

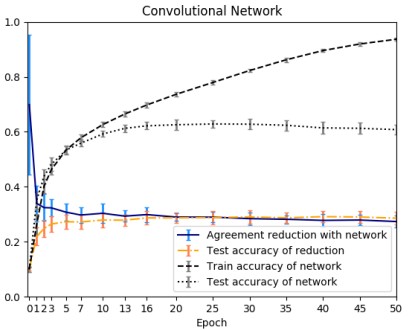 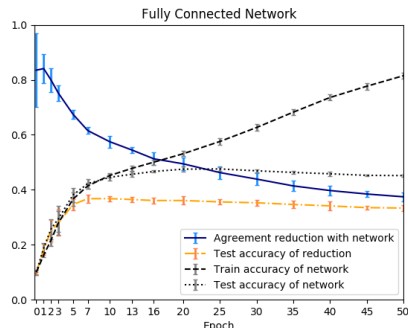

Figure 4: Mean and standard deviation of the performance of extracted tropical function during training over 5 networks. The performance of the CNN network function suffers strongly from extraction early in training, whereas the FCN shows a slow, gradual decline.

**Re-training the network** The observation of smaller angles for FCNs in Figure 2 further supports our interpretation that coefficient values of linear regions play a larger role in classification for FCNs, since smaller angles together with small Euclidean distance are explained by a similarity of the coefficient values. This suggests to also compare the similarity of linear coefficients after re-training the network in order to further understand the information encoded in the linear coefficients. Again, we find that the coefficient vectors of two separately trained CNNs are close to orthogonal, whereas both angles and distances are considerably smaller for FCNs.[6] Plots are shown in Appendix E.6.

For each dimension, we additionally compute the Pearson correlation between the linear coefficients of two separately trained networks over all training samples. We reduce the resulting vector to a single number by averag-

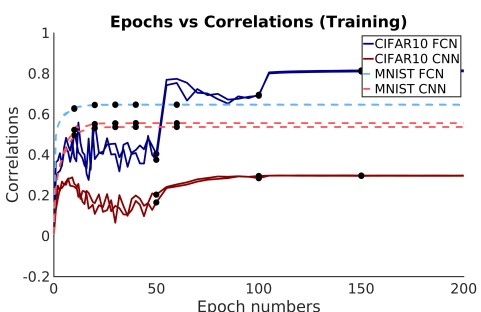

Figure 5: Pearson correlation between the linear coefficients of two separately trained networks averaged over all input dimensions. Black dots indicate a reduction of the learning rate by a factor of 10. Coefficients are more correlated after re-training for FCNs than for CNNs, suggesting that FCNs encode more information in the coefficients of linear regions than CNNs

ing the correlation factors over the dimensions. We experiment with two pairs of FCNs and CNNs trained on CIFAR10 on MNIST. Figure 5 shows the evolution of the correlation during training, confirming that the similarity of coefficient values is also larger for FCNs than for CNNs if measured by correlation. The correlation of linear coefficients after re-training and convergence for the FCNs is significant. Interestingly, for the networks trained on CIFAR10, we notice jumps in the correlation values precisely when the learning rates get decreased.

---

[6]We are comparing linear coefficients of the full network function instead of weight vectors. Whereas symmetries in the parameterization of a network function make comparisons of weight vectors complicated, the comparison of linear coefficients is well-defined.

## 5 CONCLUSION

The function of a ReLU network is piecewise linear, with an astronomically high number of linear regions. We introduced TropEx, an algorithm to systematically extract linear regions based on data points. The derivation is based on a matrix representation of tropical functions that supports efficient algorithmic development. TropEx enables investigations of the linear components of piecewise linear network functions: By extracting the networks' linear terms, the algorithm allows us to compare training and test regions and to systematically analyze their linear coefficients. Applying TropEx to fully-connected and convolutional architectures shows significant differences between linear regions of CNNs and FCNs. Other possible use cases are outlined in Appendix G. Our findings indicate a potential benefit of shifting focus from counting linear regions to an understanding of their interplay, as differences between CNNs and FCNs may be found in the coefficients of the extracted linear terms. Several measures of similarity indicate that the linear terms of CNNs are more diverse than those of FCNs and suggest that CNNs efficiently exploit the structure imposed by their architecture, whereas FCNs rely on encoding information in the values of linear coefficients.

ACKNOWLEDGEMENTS

This work was supported in part by the European Research Council Consolidator grant SEED, CNCSUEFISCDI PN-III-PCCF-2016-0180, Swedish Foundation for Strategic Research (SSF) Smart Systems Program, as well as the Wallenberg AI, Autonomous Systems and Software Program (WASP) funded by the Knut and Alice Wallenberg Foundation.

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
