# OpenReview forum: "TropEx: An Algorithm for Extracting Linear Terms in Deep Neural Networks"
_ICLR.cc/2021/Conference — ICLR 2021 Poster_

### Official Review · AnonReviewer4 · 2020-10-27
**Tropical rational map is a nice idea, but might not by capturing useful information about non-linear mappings.**

**Rating:** 6
**Confidence:** 3

**Review:**

The paper proposes a method for representing a computation of a neural network over a finite set of inputs by a projection onto a set of linear transformations (from the input space) called tropical rational map (TRM).  TRM is essentially a snapshot representing the internal representation of however deep network based on its response to a finite set of inputs.  Authors use this linearised approximation of the network internal to gather information about the complexity and generalisability of its function mapping.

The paper is well written, and well presented -  the push of most of the details on tropical algebra to the appendices is appreciated.    I think the idea of a transformation of internal representation from a deep model to a set of linear transformations has some merit, though I am not familiar with tropical algebra, and I can't completely verify all the math from appendices.  I am happy to accept that TRM can approximate network's computation over a finite set of inputs and I agree with the authors that counting linear regions alone might not tell us all that much about complexity of the mappng.  However, I am not sure there is enough in this paper to convince me that TRM evaluated on training data contains enough information to tell us something useful about generalisation capabilities of the underlying model .  Going by Table 2, and the performance of TRM-based classification on CIFAR10, it seems that perhaps overall the TRM is just a linearisation of the neural network.  That is, regardless if the mapping provided by the original network is close to linear (MNIST case) or quite non-linear (CIFAR10 case), the information captured by TRM contains only linear aspects.  I think that might be the reason why performance of TRM-based approxiation for MNIST is similar to the original network and drops from 71% to below 42% (the latter happening to be approximately the accuracy of a linear model on CIFAR10).

Figure 2 seem to provide more evidence that TRM might only capture information about simpler representation.  The difference of angles between TRM projections inside a FC network due to correctly and incorrectly classified  points are a nice result…but they seem not to capture any differentiating information about dynamics of a representationally richer architecture of a convolutional neural network.  BTW, it is not specified what dataset is Figure 2 evaluated on.

It seems to me that the whole analysis that is presented in this paper makes an assumption that the regions between the samples used to generate TRP are linear…and so the analysis tells us something about the representation when this assumption is correct (the MNIST case), but not much when the assumption is not corect (CIFAR10 and (presumably) other more complex mappings).

---

> ### Author Response · Authors · 2020-11-17
> **Reply to AnonReviewer4**
>
> We thank the reviewer for their discussion of (non-)linear mappings and their relation to our work. There are a few points speaking against the offered interpretation.
>
> **1. Clarification on linearity of functions**
> First, we would like to clarify that our reduced tropical function does not linearly interpolate between samples. There must be at least one point of nonlinearity on the connecting line between two samples (train or test).  Any network with ReLU activation is itself a tropical rational map (TRM) by [Zhang2018] composed of an extremely large number of linear functions, from which we extract another TRM where the number of linear components is equal to the size of the training data set. In that sense we agree that the extracted TRM is a linearisation of the original TRM, but it remains a ReLU network function.
>
> **2. Clarification of contributions and conclusions**
> The review focuses on the extracted TRM and its performance on test data. We emphasize that the objective of the paper is not to use the extracted TRM as an approximation to the neural network and the drop in test accuracy after extraction is not something negative, but it allows us to investigate differences between different network architectures (e.g. studying the reasons for a low drop in FCNs vs a high drop in CNNs). Our objective is to provide a tool for understanding the network’s linear regions by extracting and analysing the corresponding linear coefficients.
>
> Most importantly, the interpretation proposed by the reviewer on linearly classifiable datasets cannot explain the differences we observe between fully connected and convolutional networks. Both are reduced on the same dataset to the same type of piecewise linear function, but show significant differences. Hence, how well a dataset can be represented by linear functions can neither explain the difference in performance we observe, nor the similarity of the reduced functions after retraining or the differences in linear coefficients between training and test samples.
> These results all demonstrate how the architecture affects the information encoded in the coefficients of the reduced function. This tells us something useful about generalization performances of neural networks: CNN and FCN significantly differ in the way they use linear regions and their coefficients for generalization.
> (Figure 2 is performed on CIFAR10, which we now state. We apologize for this incompleteness.)

---

### Official Review · AnonReviewer2 · 2020-10-28
**Interesting perspective on ReLU networks**

**Rating:** 8
**Confidence:** 3

**Review:**

The paper is easy to read and technically sound. It is original and presents an interesting perspective on how to understand the activations of ReLU networks. This could be pave the way for other contributions in areas such as adversarial attacks, pruning and especially generalization theory.

The algorithm presented is efficient to the point that it can be used for large networks and datasets. And the insights on the differences between a FCN and a CNN are quite significant. It is also surprising that by the large number of linear regions of such a network, that such a high generalization can be achieved for FCNs.

I however believe that the paper would be stronger with the addition of a small experiment, namely, comparing the results from MNIST using Fashion MNIST. Clearly MNIST is too simple, but CIFAR is already too complex. Is F-MNIST an intermediate point? would it still manage to generalize or is the generalization behavior just an outlier for very simple datasets such as MNIST.

You also mention that the generalization capabilities are not explained by test points falling into the same linear regions as the training samples, but, is it possible that there are overlaps? or that the testing regions are subsets of the training ones? At least for FCNs it seems that your generalization results would indicate this is the case. Furthermore, are neighboring regions smooth on the class label? This would be significant for pruning.

Have you considered estimating the volume of the regions around the training data points? This could also provide more insights into the different behavior of FCNs and CNNs.

In general, I find the paper very interesting and it could have impact on other significant areas of research in DNNs.

---

> ### Author Response · Authors · 2020-11-17
> **Reply to AnonReviewer2**
>
> We thank the reviewer for the positive and encouraging feedback and the valuable suggestions for additional experiments that we have carried out.
>
> **1. Fashion MNIST**
> Fashion MNIST does indeed give an intermediate case with a slight, but not drastic drop in accuracy for both CNN and FCN. The drop for the CNNs is on average twice as large as the one for the FCNs, which is consistent with the other results. On average, the convolutional networks fall from a test accuracy of 89.4% to 84.7% and the fully-connected networks fall from an average 85.8% to 83.5%, producing drops of 4.7% and 2.3%, respectively. The individual results can be found in Appendix E.1, Table 4. Currently, Fashion MNIST results are based on single runs for each architecture, which will be updated to averages over five runs.
>
> **2. Overlap of linear regions**
> We can exclude the possibility of overlaps of the linear regions (maximally connected subset where the function is linear). (The extracted coefficients corresponding to all training and test regions are different and define the linear region, hence an overlap is not possible.) We agree that the experiments on the linear terms suggest that for fully connected networks there is a similarity of training and test regions.
>
> **3. Smoothness of class labels and volume of regions**
> We carried out the following experiment for two fully-connected (8 Layers and Wide) and two convolutional architectures (Conv and Narrow), showing a considerable smoothness in class labels. As a representational example, we present the results for the CNN-architecture “Conv”.
> We sampled training images X and picked a random direction d in D of length 1. We chose steps s in a range between 10^-8 and 30 (with exponential increments) and computed the activation patterns and the network label of X+s&ast;d. We recorded the smallest distance at which the activation pattern and the label changed. We did observe a few ”adversarial examples” at a distance of 0.04 (corresponding to changing each pixel by less than 1 in the range [0,255]), but overall the label assignment is smooth with an average distance of 8 (corresponding to changing each pixel by 37 in the range [0, 255]). In 11.6% of cases the label never changed. We noticed that the network assigned the same label to almost all points on the boundary of the image hypercube [0,1]^{32x32x}, explaining many of the label assignments that never change.
> The linear regions changed at a small distance, i.e., we observe a new linear region at an average/min/max distance of 10^-4/2&ast;10^-8/3&ast;10^-3. This experiment also lets us estimate the volume of the linear regions and therefore estimate the number of linear regions in the image domain. According to our calculations, Conv/Narrow/Wide/8 Layers have 10^10000/10^6700/10^5600/10^4300 linear regions inside the image domain, with corresponding test performances of 71.8/74.7/46.7/49.0.
> All results will be included in the supplementary material.

---

### Official Review · AnonReviewer1 · 2020-10-31
**Interesting study**

**Rating:** 6
**Confidence:** 3

**Review:**

This submission proposes a new method that can extract a linear region from the network function such that the region contains a specific data point. Though the problem seems to be complex, the propose method compute linear coefficients "recursively" through layers. It only needs the computation of a forward computation.

Although I hold the idea that global analysis of the function surface is more important than local analysis, I like the proposed method.

In the experimental analysis, the paper compares FCN and CNN by checking test performances, the extracted function, and coefficients. There are many interesting observations, but I expect to see some informative conclusions -- what are indications of these observations in terms of improving current methods or interpreting the model?

The submission devotes a lot of analysis to the difference between FCN and CNN. I think most of the difference is because the CNN has many more linear regions and more complex function surface, which are harder to simplify, comparing to FCN.

Is it possible to look into model and check these coefficients? For example, visualizing the coefficients in images? The visual patterns from two different networks may be very different.

Is it possible to check the local landscape of the function surface? If we add some perturbation to one example, how different coefficients can we get? With that linear region be similar to the original one? This might be useful for studying the robustness of these neural networks.

A few detailed comments:

1. The notation i is overloaded: it indexes both classes and instances.
2. The unnumbered function in 3.1 uses A matrix to represent two linear functions. Using your notation in (2), they should be different a functions.
3. More CNN architectures might make this analysis more interesting. For example, the residual links in a ResNet may have some relation with these linear coefficients. Also the analysis of a CNN on cifar10 is more meaningful when it has higher performance.

---

> ### Author Response · Authors · 2020-11-17
> **Reply to AnonReviewer1 - Part 2**
>
> **5. On the detailed comments**
> a) Overloaded notation
> We understand that the concern about the overloaded notation refers to the notation used in Theorem 3.3 and equation (2) to describe the dataset, which we have now changed.
> b) Unnumbered function in 3.1
> We cannot follow the issue on the unnumbered function to represent two linear functions. The difference in (2) is represented by the tropical quotient in 3.1, and Lemma 3.2 shows that the matrix-vector pair in the denominator can be chosen to be constant. We would appreciate a clarification to improve the presentation if something is still left unclear.
> c) ResNet
> Results for a ResNet architecture will be added within the next few days.

---

> > ### Comment · AnonReviewer1 · 2020-11-23
> > **A quick note**
> >
> > To be clear: instead of asking for new results for this review, the review suggests directions of improving this submission. So please do not feel pressured to generate new results.

---

> ### Author Response · Authors · 2020-11-17
> **Reply to AnonReviewer1 - Part 1**
>
> We thank the reviewer for the positive evaluation and good suggestions for additional experiments most of which we have carried out.
>
>
> **1. Informative conclusions on interpreting the model**
> Apart from deriving a non-trivial algorithm, thereby contributing a tool for a new type of investigation of ReLU networks, we were able to draw the following conclusions from our experiments:
> a) We showed that CNNs and fully connected networks have different mechanisms of using linear coefficients and linear regions to generalize to unseen data (similarity of the linear coefficients is a large factor in the classification of test samples and re-training leads to similar coefficients for FCNs, but not CNNs). A theoretical explanation of the generalization capabilities of neural networks must take this difference into account. Similarly, these observations could be of relevance for understanding and improving pruning techniques and adversarial attacks.
> b) Neither the number of linear regions alone seems to be a good indicator, nor the individual linear coefficients for CNNs, whereas it is the interplay of linear regions that needs to be understood. There have been significant research efforts [Pascanu2013, Montufar2014, Arora2016, Montufar2017, Ragu 2017, Serra2018, Zhang2018, Hanin&Rolnick2019b, Xiong2020] devoted to obtaining estimates of the maximal number of linear regions. Hence, our results can have a positive effect of directing research to different questions.
> c) Training and test samples fall into different linear regions. Test regions are also significantly different from neighboring regions, in particular for CNNs.
> d) MNIST can be classified correctly with few linear regions - this observation does not hold for CIFAR10. This demonstrates the importance to study linear regions on sufficiently complex datasets, which is computationally hard for some approaches to linear regions.
>
> **2. On the number of linear regions of CNNs**
> The reviewer suggests that the simple reason for our observations could be that CNNs have many more linear regions. From the perspective of calculating the maximal number of linear regions that FCN and CNN can generate, the opposite holds. That is, we compare two networks that have the same number of neurons in each hidden layer (not the same number of parameters), where one consists of fully connected layers and the other one of convolutional layers. Fixing the number of nodes after each layer, the class of CNNs is a strict subset of the class of FCNs (since we can obtain a CNN by setting all parameters not corresponding to convolutional filters equal to zero and enforce suitable weight sharing). Hence, the maximal number of linear regions over all parameter configurations for FCNs must be greater or equal to the one for CNNs. This shows that the maximal number of linear regions is not a good measure.
>
> By computing the distance to neighbouring activation patterns, we now estimate the number of linear regions in the image domain ([0, 1]-hypercube). (Experimental details in reply to AnonReviewer2 under ‘3’). We find that in practice, the CNN does indeed have more linear regions than the fully-connected network (10^6700 vs 10^5600). However, the number of linear regions can still not explain performance differences between our networks: For example, the performance of Narrow (10^6700 regions) is better, but Conv has much more linear regions (10^10,000 regions). We observe the same for FCN - Wide and FCN - 8 Layers: Here 8 Layers (10^5600 regions) performs better, but Wide (10^4300 regions) has more linear regions. We are going to integrate these results into the paper within the next few days.
> The lack of a simple explanation from counting the number of linear regions underlines the impact of our work. We also hope that the reviewer can appreciate our results as our experimental evidence is in line with their opinion that a local analysis of the loss cannot be sufficient.
>
> **3. Visualising linear coefficients**
> We performed an additional experiment visualizing the coefficients (Supplements E.7). We observe differences between the CNN and the FCN, where one might be able to recognize the class labels from the FCN in a few cases, but we find the results too weak to draw conclusions.
>
> **4. Changes of coefficients when perturbing inputs**
> We checked the robustness of the network by sampling different directions and recording at what distance the activation changes (i.e. neighboring linear region) and at what distance the label assignment changes (see details in our reply to AnonReviewer2), showing a considerable robustness of label assignment to neighboring regions. We will report the similarity of the coefficients after perturbing inputs in the following days.
>
> Reply to be continued in Part 2.

---

> > ### Comment · AnonReviewer1 · 2020-11-24
> > **Some further comments**
> >
> > Thank you for detailed explanation. However, I still feel that that there are not many definite conclusions from this work. For example, a) CNN and fully connected networks uses linear regions differently, is somewhat expected. It is similar for c) and d).
> >
> > I feel that individual linear regions are not key factors of a learner. I have to arguments: 1) If we use tanh as activations, we are likely to get similar classification results. 2) a test data point falls in a linear region containing no training points, then the classifier still have a good chance to predict correctly. Therefore, I feel the global function surface or at lease surface forming some meaningful structure in a big region is the key to understand behaviors of these models.
> >
> > Finally, I want to say that the method in this work is still interesting. The study of linear regions might be useful in ways I don't know (my previous point may be biased due to the limitation of my knowledge). This is the basis of my recommendation of this submission.

---

> > > ### Author Response · Authors · 2020-11-24
> > > **Thank you for the additional comments!**
> > >
> > > We again thank the reviewer for the thoughtful further comments and the quick note. We highly appreciated the suggestions and agreed that including additional results could strengthen our point. While we respect the reviewer's opinion that linear regions and their coefficients in general may be misleading to study, we would like to quickly respond to the observations from our point of view.
> > > 1) Tanh and ReLU activations give rise to very different functions that require different means to study them.  Although tanh() activations can lead to similar performance, linear regions may still encode essential information about the learner, justifying the study of linear regions as a standard approach.
> > > 2) The fact that test regions work well means that there is a (global) flow of information from training to test regions and our results suggest that this flow of information must be quite different for CNNs und FCNs, although both network functions belong to the class of hierarchically composed piecewise linear functions.

---

### Official Review · AnonReviewer3 · 2020-11-01
**TropEx: An Algorithm for Extracting Linear Terms in Deep Neural Networks**

**Rating:** 6
**Confidence:** 3

**Review:**

Summary: This paper studies the role of linear terms in the network performance using nontrivial tropical algebra inspired algorithms. In particular, the paper extracts linear terms associated with the linear regions of only the training points, and uses this to generate an extracted network function. The paper proposes an algorithm TropEx to systematically extract linear terms from piecewise linear network functions built using activation functions such as ReLUs. It is shown that this extracted network can be used for classification on the test data as well. In fully connected networks, such a modeling seems to work well on the test data. In the case of CNNs, the gap between the extracted network and original one is large. The paper argues that the number of linear regions may not be the right metric for expressiveness and generalization.


Pros:

1) The network function can be represented as the difference between maxima of a really large set of affine functions on the input data as shown in Equation (1). The paper nicely shows that in piecewise linear functions learned using a finite set of training samples, we generate exponential numbers of linear regions. The training data typically falls in a very tiny fraction of these regions, and only these linear regions are important. The general study of the number of linear regions and bounds are driven by quantifying expressiveness of neural networks. This work takes a different approach and attempts to only consider the linear regions associated with the training samples.


2) The paper extracts linear terms associated with the linear regions of only the training points, and uses this to generate an extracted network function. The extracted network function N^(x) and the original network N are shown to agree in the neighborhood of the training samples with respect to the assigned labels.


3) The paper nicely illustrates that all training and test points fall in different linear regions.


Cons:


1) While the general motivation is nice, the paper makes some strong claims without enough justification. In the list of contributions, the paper claims that the number of linear regions is not a good indicator for network performance. However, there is no formal result of empirical validation that justifies this. The proposed approach to extract network function based on the training data is heavily influenced by the linear regions associated with the training data.


2) The paper does not clearly explain what “linear terms” mean and it would be useful to define them formally. Does it refer to the affine functions or the linear coefficients w’s?


3) Figure 2 requires some clarification. We take a test sample and compute the coefficients of the affine functions in the original and the extracted function. We compute the difference using Euclidean and angular distance. For CNNs, the coefficient vectors are orthogonal for both correct and incorrectly classified points. In the case of FCNs, the distance of correctly classified points is smaller than for incorrectly classified points. This basically means that the generalization is better in FCN compared to CNNs.


4) Table 2 shows that the gap between N and N^(x) is large for CNNs. If the extracted network can produce correct labels for all the training data, it is hard to see why the results are poor for the test data. Considering the large amount of training data, even nearest neighbor methods can perform reasonably well.

---

> ### Author Response · Authors · 2020-11-17
> **Reply to AnonReviewer3**
>
> **1. On linear regions being an indicator of network performance**
>  	 We thank the reviewer for pointing us to parts that may need clarification. In particular, we would like to clarify why we provided sufficient evidence to confirm that the number of linear regions is not a good indicator for neural network performance. Our conclusion is not only based on the fact that the extracted functions share the same number of linear regions and show differences in their performance, but we presented more evidence:
>  	Previous research focused on obtaining an estimate of the maximal number of linear regions over all parameter configurations to measure the expressiveness of the network. Fixing the number of nodes after each layer, the class of CNNs is a strict subset of the class of FCNs (since we can obtain a CNN by setting all parameters not corresponding to convolutional filters equal to zero). Hence, the maximal number of linear regions for FCNs must be greater or equal to the one for CNNs. In our experiments the FCN “Wide” and the CNN “Narrow” share the same number of nodes after each layer (see page 7, second paragraph). Since “Narrow” performs better on test data than “Wide”, but the maximal number of linear regions is higher for “Wide”, it follows that this number alone is not a good indicator of network performance. We will attempt to make this more clear.
> Based on experiments suggested by AnonReviewer1 and AnonReviewer2, we now provide additional numerical evidence, where we estimate the number of linear regions for a trained CNN and a trained FCN . According to our calculations, Conv/Narrow/Wide/8 Layers have 10^10000/10^6700/10^5600/10^4300 linear regions inside the image domain, with corresponding test performances of 71.8/74.7/46.7/49.0. Despite having less linear regions, Narrow performs better than Conv, and 8 Layers performs better than Wide, further supporting that we cannot draw conclusions on the performance from the number of linear regions.
>
> **2. Definition of linear terms**
> We thank the reviewer for suggesting to state the definition of linear terms
> more clearly and to distinguish affine functions from their linear coefficients. The term “linear term” shall denote an affine function and we now only use the terminology in this context.
>
> **3. Clarification of Figure 2**
> We cannot follow the conclusion that generalization would be better in FCNs compared to CNNs from the experiment of Figure 2. From this figure, we can only infer that for FCNs, correct classification is correlated with similar coefficients applied to test and training samples, while this is not the case for CNNs. Do we understand correctly that the reviewer assumes that in CNNs correct generalization requires similarities in the coefficients since we observe this correlation for FCNs? Our observations reveal that this assumption would be incorrect and that CNN and FCN use the linear terms differently for generalization.
>
> **4. Comparison to nearest neighbours**
> Using nearest neighbours on CIFAR10 gives a test accuracy of slightly above 30%, so the extracted function from the CNN indeed lands in the range of k-nearest neighbor classification and for k=1 it trivially holds that 1-NN similarly correctly classifies all training samples. This supports our argument that the CNN uses the architecture efficiently for generalization, while its linear coefficients of training regions are limited in meaning.
> We do not agree that low accuracies of the extracted functions for CNNs form a weakness of our paper. Our goal is not to use these extracted functions for classification in practice, but to perform a theoretical analysis of linear regions and their coefficients.

---

### Decision · Program_Chairs · 2021-01-07
**Final Decision**

**Decision:**

Accept (Poster)

**Comment:**

This paper analyses linear regions in ReLU networks using a new algorithm for extracting linear terms based on the data. The reviewers found the paper to be well written with sound results. While the paper itself provides only modest evidence of the algorithm’s utility (mainly in terms of highlighting some distinctions between fully-connected and convolutional networks), the algorithm and the corresponding new paradigm of exploring linear terms rather than counting regions may prove useful in future analyses. Altogether, I think this paper will interest theorists focusing on ReLU networks and I recommend acceptance.